# The Expression Pattern of tRNA-Derived Small RNAs in Adult *Drosophila* and the Function of *tRF-Trp-CCA-014-H3C4* Network Analysis

**DOI:** 10.3390/ijms24076169

**Published:** 2023-03-24

**Authors:** Deying Yang, Feng Xiao, Ya Yuan, Jiamei Li, Siqi Wang, Xiaolan Fan, Qingyong Ni, Yan Li, Mingwang Zhang, Xiaobin Gu, Taiming Yan, Mingyao Yang, Zhi He

**Affiliations:** 1College of Animal Science and Technology, Sichuan Agricultural University, Chengdu 611130, China; deyingyang@sicau.edu.cn (D.Y.); 2Farm Animal Genetic Resources Exploration and Innovation Key Laboratory of Sichuan Province, Sichuan Agricultural University, Chengdu 611130, China; 3College of Veterinary Medicine, Sichuan Agricultural University, Chengdu 611130, China

**Keywords:** tRNA-derived small RNAs (tsRNAs), expression patterns, *tRF-Trp-CCA-014*, *H3C4*, *Drosophila*, mouse NIH3T3 cells

## Abstract

tRNA-derived small RNAs (tsRNAs) are derived from tRNA and include tRNA halves (tiRNAs) and tRNA fragments (tRFs). tsRNAs have been implicated in a variety of important biological functions, such as cell growth, transcriptional regulation, and apoptosis. Emerging evidence has shown that Ago1-guided and Ago2-guided tsRNAs are expressed at 3 and 30 days in *Drosophila* and that tRF biogenesis in fruit flies affects tRNA processing and tRNA methylation. However, a wide analysis of tsRNA patterns in different ages of *Drosophila* have not been reported via the small RNA sequencing method. In the present study, tsRNAs of young (7 days) and old (42 days) *Drosophila* were sequenced and their expression characteristics were analysed. Then, a specific tRF (named *tRF-Trp-CCA-014*) was determined and was found to be conserved in fruit flies, mice, and humans. The expression patterns of *tRF-Trp-CCA-014* in different tissues and stages of fruit flies and mice, and mouse NIH/3T3 cells were detected. Furthermore, mouse embryonic fibroblast NIH/3T3 cells were used as a model to analyse the function and targets of *tRF-Trp-CCA-014*. The RNA-seq data of six groups (Mimics, Mimic NC, Inhibitors, Inhibitor NC, Aging (adriamycin), and Control (Normal)) in mouse NIH3T3 cells were analysed. The results showed that the number of tsRNAs at 42 days (417) was more than at 7 days (288); thus, it was enriched with age. tRFs-1 were the most enriched, followed by 5′-tRFs and 3′-tRFs. Twenty-one differentially expressed tsRNAs were identified between 7 days and 42 days. Then, the conserved tRF *tRF-Trp-CCA-014* was identified and found to accumulate in aged fruit flies and aged mouse NIH3T3 cells. RNA-seq data showed that most differentially expressed genes were involved in the immune system, cancer: overview, and signal translation. Furthermore, *tRF-Trp-CCA-014* was found to bind to the 3′UTR of *H3C4* in a dual-luciferase reporter gene assay. *tRF-Trp-CCA-014* and *H3C4* were detected in the cytoplasm of aged NIH3T3 cells by RNA in situ hybridization. These results suggest that the *H3C4* gene is the target of *tRF-Trp-CCA-014*. This study will advance the current understanding of tRF roles and their implication in *Drosophila* and mouse studies.

## 1. Introduction

tRNA-derived small RNAs (tsRNAs) are a class of noncoding RNAs with important biological functions and include two major types: tRNA-derived fragments (tRFs) and tRNA halves (tiRNAs) [1,2]. Generally, tsRNAs have various mechanisms in biological processes, such as interacting with proteins to regulate gene expression [3], targeting genes to inhibit cancer cell proliferation and affect early neural development [4,5], and being involved in posttranscriptional methylation modification of tRNA [6]. To date, tsRNAs have been found to be expressed widely across a range of organisms: humans [4], *Danio rerio* [7], *Caenorhabditis elegans* [8], mice [9], *Saccharomyces cerevisiae* [10], and *Drosophila melanogaster* [11].

Based on the mapped positions of tRFs to tRNAs, there are four types: tRF-1, tRF-2, tRF-3, and tRF-5 [1,12,13,14]. tRF-1 (14–33 nt) is generated from the 3′ ends of the precursor tRNA. The 5′ end of tRF-1 matches the cut site of RNase Z, and the 3′ end matches an RNA polymerase III (RNA pol III) transcription termination signal. tRF-2 includes only the anticodon stem and loop tRNA. tRF-3 is generated from the 3′ ends of the mature tRNA and subclassified as tRF-3a (17–18 nt) and tRF-3b (19–22 nt). Furthermore, tRF-5 is produced from the 5′ ends of the mature tRNA and is subclassified as tRF-5a (14–16 nt), tRF-5b (22–24 nt), and tRF-5c (28–32 nt). tiRNAs (31–40 nt) are derived from specific cleavage in the anticodon loops of mature tRNA and classified as 5′-tiRNA or 3′-tiRNA. 5′-tiRNA begins from the 5′ end of mature tRNA and ends at the end of the anticodon loop, whereas 3′-tiRNA starts at the anticodon loop and ends at the 3′ end of the mature tRNA.

*Drosophila* is an important classic model organism for the study of human diseases and behaviours [15]. Recent studies have reported that tRFs bind to Argonaute (Ago) to function as miRNAs by combining the 3′ untranslated region (3′ UTR) sequence of the target gene mRNA [4,16]. In previous studies, the abundance of tRFs in complex with Ago1 and Ago2 was analysed to identify Ago1-guided and Ago2-guided fragments at 3 and 30 days in *Drosophila* [11,17]. However, that analysis had limitations and could not fully capture the expression profile of tRFs in *Drosophila* because tRFs have other mechanisms in addition to miRNA-like roles, such as binding proteins. Furthermore, the expression patterns of tRFs in *Drosophila* embryos and S2 cells have been determined and a small number of tRFs have been identified [18,19]. Thus, systematic and comprehensive research should be carried out at different ages in adult *Drosophila*.

In this study, tsRNAs of young (7 days old, 7 days) and old (42 days old, 42 days) *Drosophila* were sequenced and their expression characteristics were analysed. Then, a novel tRF was named *tRF-Trp-CCA-014* and was significantly upregulated at 42 days in flies (it is conserved in *Drosophila*, mouse, and human). The transcriptomes of NIH/3T3 mouse embryonic fibroblasts were analysed after transfection of *tRF-Trp-CCA-014* mimics (Mimics), *tRF-Trp-CCA-014* inhibitor (Inhibitors), the negative control of *tRF-Trp-CCA-014* mimics (Mimic NC), the negative control of *tRF-Trp-CCA-014* inhibitor (Inhibitor NC), and in adriamycin-treated cells (aged cells, Aging (ADR)) and the corresponding control without ADR treating ((Control (Normal)), respectively. The results showed that *tRF-Trp-CCA-014* had binding activity with the *H3C4* gene. The results will advance the understanding of the expression pattern of tsRNAs in adult *Drosophila* and the function of *tRF-Trp-CCA-014*.

## 2. Results

### 2.1. Expression Pattern of Drosophila tsRNAs

For ten samples (five biological replicates for 7 days and 42 days, respectively), 64.6 M clean reads were obtained (NCBI SRA_PRJNA716466). The Q30 values ranged from 89.95–93.05%. A total of 487 tsRNAs (449 newly discovered and 38 known, Figure 1A) were identified, including tRF-1, tRF-2, tRF-3a, tRF-3b, tRF-5a, tRF-5b, tRF-5c, and tiRNA-5 (Dataset S1). Then, tRF-Asp-GTC (36 tRFs) and tRF-Glu-CTC (36 tRFs) were the most abundant types, followed by tRF-Gln-CTG (27 tRFs) and tRF-Val-CAC (25 tRFs). A total of 288 and 417 tsRNAs were identified at 7 days and 42 days, respectively. A total of 218 common tsRNAs were expressed at 7 days and 42 days. Among them, tRF-1, tRF-5a, tRF-5c, and tRF-3a were the main types (Figure 1B,C).

tRFs-1 had the highest abundance, followed by 3′-tRFs and 5′-tRFs. The length distribution of tsRNAs had similar patterns at 7 days (Figure 1D) and 42 days (Figure 1E). tRF-1 was distributed from 14 nt to 32 nt and focused on 19–21 nt and 25–27 nt. The length of tRF-2 was 14–15 nt. tRF-5a, tRF-5b, and tRF-5c were focused on 14–17 nt, 22–25 nt, and 28–32 nt, respectively. tRF-3a and tRF-3b were focused on 19–22 nt and 17–18 nt, respectively. In addition, tiRNA-5 was focused on 33–34 nt. However, there was a slight difference between 7 days and 42 days. The frequency of tRF-1 with 25 nt in 7 days was more than that in 42 days.

According to a fold change ≥1.5 and *p* ≤ 0.05, 18 tRFs were upregulated and 3 tsRNAs (1 tiRNA and 2 tRFs) were downregulated at 42 days compared with 7 days (Figure 1F,G, Dataset S1). In addition, 466 tRFs and tiRNAs were not significantly different between 7 days and 42 days. There were 18 upregulated tRFs, including 7 tRF-1, 2 tRF-3a, 2 tRF-5a, and 7 tRF-5b. Among those tRFs, 9 non-annotated and nine annotated tRFs were identified. Five tRFs (*tRF-Ala-TGC-004*, *tRF-Ala-TGC-005*, *tRF-Ala-TGC-006*, *tRF-Trp-CCA-014*, and *tRF-Val-CAC-011*) were conserved in mouse and fly. Specifically, *tRF-Trp-CCA-014* was conserved in fly, mouse, human, fish, and *Xenopus*. Furthermore, there were three downregulated tRFs, *tiRNA-Pro-CGG-001*, *tRF-Cys-GCA-014*, and *tRF-Val-CAC-016*.

### 2.2. Verification of Differentially Expressed tsRNAs

Because of tsRNA accumulation at 42 days (417 tsRNAs), this study focused on the upregulated tsRNAs at 42 days. Ten upregulated tRFs were chosen to test their expression by realtime fluorescence quantitative PCR (qPCR) (Figure 2A). The difference between the *tRF-Val-TAC-003* and the *tRF-Val-TAC-004* sequences was only one base and they were regarded as one gene if they could not be distinguished. The same situation was true for *tRF-Ala-AGC-007* and *tRF-Ala-AGC-008*. The results show that the expression patterns of *tRF-Ala-AGC-007/008*, *tRF-Val-TAC-003/004*, *tRF-Trp-CCA-014*, *tRF-Val-CAC-011*, *tRF-Ala-TGC-004*, *tRF-Ala-TGC-005*, and *tRF-Ala-TGC-006* by qPCR were consistent with the RNA-seq results (Figure 2B). There was no significant difference in the expression levels of *tRF-Val-CAC-020* at 7 days and 42 days. Significantly, *tRF-Trp-CCA-014* had the highest expression among these tRFs.

ADR is an anthracycline antineoplastic drug widely used to construct a model of mouse cell senescence [20]. The β-galactosidase activity and pro-inflammatory cytokines *IL-1* and *IL-6* are consistently present in the senescence-associated secretory phenotype (SASP) [21,22,23]. The β-gal staining (Figure 2C–E) and the expression levels of *IL-1α*, *IL-1β*, and *IL-6* (Figure 2F) increased in aged mouse NIH3T3 cells. The above results show that the aged mouse NIH3T3 cell model was successfully built using ADR. Then, *tRF-Trp-CCA-014* was quantified and found to be significantly higher in senescent cells than in normal cells (Figure 2G). Thus, *tRF-Trp-CCA-014* was used as the subject of follow-up studies.

### 2.3. Expression Features of tRF-Trp-CCA-014

Based on bioinformatic analysis, *tRF-Trp-CCA-014* (tRF-3a type, 17 bp) is conserved in *Drosophila*, mouse, human, zebrafish, and *Xenopus tropicalis* (Figure 3A). The length of *tRF-Trp-CCA-014* was 17 bp in *Drosophila* and 18 bp in the other four species. The *tRF-Trp-CCA-014* of *Drosophila* was derived from tRNA-Trp-CCA-1-1 and tRNA-Trp-CCA-2-1 in chr.2R (chromosome 2R) and the *tRF-Trp-CCA-014* of mice was derived from chr.11 tRNA 1849-Trp-CCA, chr.11 tRNA 1820-Trp-CCA, chr.11 tRNA 398-Trp-CCA, chr.13 tRNA 109-Trp-CCA, chr.10 tRNA 857-Trp-CCA, chr.1 tRNA 107-Trp-CCA, and chr.10 tRNA 1282-Trp-CCA. Human *tRF-Trp-CCA-014* was derived from chr.7 tRNA 1-Trp-CCA, chr.12 tRNA 6-Trp-CCA, chr.17 tRNA6-Trp-CCA, chr.6 tRNA 168-Trp-CCA, chr.6 tRNA170-Trp-CCA, and chr.17 tRNA39-Trp-CCA.

The expression patterns of *tRF-Trp-CCA-014* in different developmental stages and tissues (42 days) were quantified. Throughout the developmental cycle, *tRF-Trp-CCA-014* was mainly expressed in the third larval and pupal stages (Figure 3B). The expression level of *tRF-Trp-CCA-014* began to increase at 42 days and the accumulation at 49 days was significantly higher than that at 42 days in the adult stage (Figure 3C). Its expression level was significantly higher in the head and fat body than in the gut and ovary of *Drosophila* (Figure 3D). Furthermore, *tRF-Trp-CCA-014* was mainly expressed in the cytoplasm of mouse NIH3T3 cells (Figure 3E). The expression level of *tRF-Trp-CCA-014* in tissues from 5-month- and 9-month-old young mice was detected; it was upregulated in the thigh muscle, the liver, and the brain of 19-month-old mice when compared with 5-month-old young mice (Figure 3F).

### 2.4. RNA-seq of Mouse NIH3T3 Cells

The mimics and inhibitors of *tRF-Trp-CCA-014* were successfully transfected into mouse NIH3T3 cells to overexpress and knock down the gene expression level, respectively (Figure 4A). The results showed that *tRF-Trp-CCA-014* cannot induce NIH3T3 cell senescence (Appendix A). The β-gal staining and expression level of p16 protein were not changed after transfection of the mimics and inhibitors of *tRF-Trp-CCA-014* (Appendix A). Then, transcriptomes were analysed for six groups (Mimics, Mimic NC, Inhibitors, Inhibitor NC, Aging (ADR), and Control (Normal)) by RNA-seq. A total of 123.62 G of clean data were obtained (Dataset S2). The effective data volume of each sample was 6.03–7.28 G (NCBI SRA_PRJNA934729). The Q30 base was distributed in 93.23–95.41% of the transcriptome and the average GC content was 51.21%. By aligning the reads to the reference genome, the alignment rate was 87.66–92.40%. The fragments per kilobase million, fragments per kilobase of exon model per million mapped fragments (FPKM) value patterns of different groups were similar (Figure 4B).

Based on *p* < 0.05 and f|log2FC| > 1, differentially expressed genes (DEGs) were identified (Dataset S3). There were 33 (17 upregulated and 16 downregulated in the Mimics group, Figure 4C), 104 (101 upregulated and 3 downregulated in the Inhibitors group, Figure 4D), and 172 (39 upregulated and 133 downregulated in the Mimics group, Figure 4E) DEGs among the Mimics vs. Mimic NC groups, Inhibitors vs. Inhibitor NC groups, and Mimics vs. Inhibitors groups, respectively. The most DEGs, 5316 were identified between the ADR treatment group (ageing) and the Control group (normal).

### 2.5. KEGG and GO Enrichment of DEGs

KEGG and GO enrichment analyses of the identified DEGs were performed. The overlapping KEGG signalling pathways (Dataset S4) were enriched in cellular processes, environmental information, genetic information processing, human diseases, metabolism, and organismal systems among Mimics vs. Inhibitors groups (Figure 5A), Mimics vs. Mimic NC groups (Figure 5B), and Inhibitors vs. Inhibitor NC groups (Figure 5C). The signalling pathways associated with the most DEGs focused on the immune system, cancer: overview, and signal translation. Furthermore, the results showed that most DEGs were enriched in similar biological processes in GO analysis (Dataset S3) among Mimics vs. Inhibitors groups (Figure 6A), Inhibitors vs. Inhibitor NC groups (Figure 6B), and Mimics vs. Mimic NC groups (Figure 6C). The most enriched biological process terms included “biological regulation”, “cellular process”, “metabolic process”, “regulation of biological process”, and “signalling”; the cellular component terms included “macromolecular complex”, “membrane”, “membrane-enclosed lumen”, and “organelle”; the molecular function terms included “binding”, “catalytic activity”, “molecular transducer activity”, and “transporter activity”.

### 2.6. Analysis of tRF-Trp-CCA-014 Targets

Targets of *tRF-Trp-CCA-014* were screened through two methods based on the DEGs from the RNA-seq data of mouse NIH3T3 cells.

The functional DEGs were utilized for screening. First, 129 functional DEGs were identified from the overlapping genes between 5316 DEGs (Aging (ADR) vs. Control (Normal)) and 172 DEGs (Mimics vs. Inhibitors) (Figure 7A). Second, 26 common functional DEGs were enriched in immune system, cancer: overview, and immune disease (Figure 7B). Finally, 14 functional DEGs were identified in the intersection of two sets (129 common DEGs vs. 26 DEGs), including *H4C11*, *H4C1*, *H4C18*, *Fcgr1*, *H4C8*, *H2bc6*, *H2bc18*, *H3C4*, *H2bc4*, *H3C11*, *H4C12*, *H3C2*, *H4C9*, and *H4C14* (Figure 7C).

Additionally, the DEGs of transcription factor families were used to select the targets of *tRF-Trp-CCA-014*. Previous studies reported that tsRNAs are involved in DNA-binding transcription factor activity and protein binding in endometriosis patients [24]. In the present study, by comparing the DEGs of the transcription factor families of the Mimics vs. Inhibitors groups, 11 transcription factor genes of six families had significant changes in the Inhibitors group, including 2 upregulated genes (*Pdx1* and *Batf2*) and 9 downregulated genes (Figure 7D).

Finally, 16 targets were selected randomly for verification by qPCR, including *H3C11*, *H4C11*, *H4C18*, *Dbp*, *Klf10*, and *Nr1d1*. The results showed that the expression levels of eight targets (*Klf10*, *Nr1d1*, *H4C8*, *H2bc18*, *H3C4*, *H2bc4*, *Id3*, and *Zfp97*) were consistent with the RNA-seq results (Figure 7E). In addition, the *H3C4* gene was conserved in humans, mice, and *Drosophila,* with 90.51% sequence similarity (Figure 7F). Thus, the targeting relationship between *tRF-Trp-CCA-014* and *H3C4* was selected for further study.

### 2.7. Analysis of the tRF-Trp-CCA-014-H3C4 Network

The binding sites between *tRF-Trp-CCA-014* and *H3C4* of mouse were predicted. The results showed that there were the binding sites between *tRF-Trp-CCA-014* and the 3′UTR of *H3C4* based on *p* < 0.001 (Figure 8A). Then, a mutant of *H3C4* was designed by mutating CCCC to TTTT (Figure 8B). Furthermore, the binding sites were verified by a double luciferase reporter gene assay in HEK-293T cells. The luciferase activity was significantly decreased after transfection with wild-type (WT) *tRF-Trp-CCA-014* mimics. 3′UTR of *the H3C4* plasmid was compared to transfection with the negative control of *tRF-Trp-CCA-014* and WT of the *H3C4* 3′UTR plasmid group, the mutants of the *H3C4* 3′UTR plasmid and the mimics of the *tRF-Trp-CCA-014* group, and the mutants of the *H3C4* 3′UTR plasmid and the negative control of the *tRF-Trp-CCA-014* group, separately (Figure 8C). The expression level of H3C4 protein was also determined in mouse NIH3T3 cells. The mimics of *tRF-Trp-CCA-014* increased H3C4 protein levels, while the inhibitors of *tRF-Trp-CCA-014* decreased H3C4 protein levels (Figure 8D,E). The location of *tRF-Trp-CCA-014* and *H3C4* was determined by RNA in situ hybridization in aged mouse NIH3T3 cells (Figure 8F–K). The results showed that the fluorescence signals of *tRF-Trp-CCA-014* and *H3C4* were found in the cytoplasm of aged mouse NIH3T3 cells.

## 3. Discussion

tsRNAs are found in a wide variety of organisms and tissues with potential utility as biomarkers and therapeutic targets and are involved in various cellular and molecular processes, including cell growth, differentiation, and transcriptional regulation [25]. The expression patterns of tRFs in several organisms have been reported. The expression level of 3’-tRFs increased with age (6 months, 14 months, and 22 months) based on miRNA data from the mouse brain [26,27]. Similar results were observed in *C. elegans* [8,28]. The tRFs of zebra fish are differentially expressed during development and in differentiated tissues, suggesting that they are likely produced by specific processing [7]. These results suggest that tsRNAs are enriched in aged organisms and it is essential to widely study tsRNAs at different ages in *Drosophila*.

Studies have shown that tRFs are expressed in *Drosophila* [11,25]. Karaiskos et al. identified the tRFs of fruit flies into RISC complexes according to Ago1 and Ago2 coimmunoprecipitated libraries [11]. They found that the total levels of Ago-loaded tRFs changed with age and that most age-related tRFs were enriched in Ago2-loaded situations [11]. These results suggest that tRFs have miRNA-like functions in *Drosophila*. Although Ago-loaded tRFs were identified, they may represent only a subset of the overall expressed tsRNAs because tsRNAs have other mechanisms in addition to miRNA-like patterns, such as acting on functional proteins [29] and transcription factors [30]. In our study, a comprehensive and systematic study of tsRNAs in 7-day- and 42-day-old *Drosophila* was performed. The number of tsRNAs at 42 days (417) was more than at 7 days (288) and tsRNAs were enriched with age. The most abundant tRF types (tRF-Asp-GTC and tRF-Glu-CTC) are consistent with the results of a previous study [11]. Then, 111 tRNA genes were identified in 0–1 and 7–8 h *Drosophila* embryos [18]. A similar pattern of the sizes of the most abundant tRFs (tRF-1) was found: 28 bp and 26-27 bp in embryos and adults, respectively. Specifically, the expression levels of tRFs changed temporally following the maternal-to-zygotic transition in embryos [18]. These results suggest that tsRNAs may have a wide range of functions in embryos and adult fruit flies.

Small RNA sequencing datasets extracted from wild-type and mutant ovaries were used to analyse the expression patterns of tRFs and two different highly conserved steps of tRNA biogenesis in *Drosophila* [25]. Both 5′-tRF and 3′-tRF are generated from a given mature tRNA [31]. A meta-analysis of more than 50 small RNA sequencing libraries demonstrated that the abundance of 5′- and 3′-tRFs from the same tRNA origin was unequal [32]. In the present study, our findings are consistent with the results of previous research [32] that tRFs-1 are the most enriched, followed by 5′-tRFs and 3′-tRFs. This consistency may be due to the method of library preparation or sequencing with standard small RNA-Seq protocols. Because tRFs-1 are poorly post-transcriptionally modified, tRFs-1 are preferential tRFs [25]. In contrast, tRFs, mainly from the 5′ ends of tRNAs in 0–1 and 7–8 h *Drosophila* embryos, co-sedimented with the non-polysomal fractions [18]. The difference in expression patterns may be related to the degradation orientation of tRNA and tsRNAs at different developmental stages. Thus, the expression patterns of tsRNAs in each developmental stage should be determined in future studies. In a previous study, according to arbitrary thresholds for illustrative purposes, 8 Ago1-associated tRFs were downregulated, 4 Ago1-associated tRFs were upregulated, and 40 Ago2-associated tRFs were upregulated in aged fruit flies [11]. Then, 13 differentially expressed tRFs were screened (*p* < 0.01 and fold changes ≥ 2) in mice [33]. In our study, 21 differentially expressed tsRNAs were identified based on fold change ≥ 1.5 and *p* ≤ 0.05. The expression patterns of tsRNAs demonstrate that they may play important roles in *Drosophila*.

To date, the function and the mechanism of specific tsRNAs present specific challenges and their biofunction is not well understood in *Drosophila*. In our study, the conserved tRF *tRF-Trp-CCA-014* was identified and enriched in aged fruit flies and aged mouse NIH3T3 cells. The gene expression regulation system (UAS/GAL4) is an effective tool to construct transgenic *Drosophila* to overexpress or knock down targets in *Drosophila* [34]. Thus, we tried to use the UAS/GAL4 system to overexpress the sequence of *tRF-Trp-CCA-014* in *Drosophila*, however, it failed. This suggests that we should pay more attention to studying their cutting mode and modification, which could help us to overexpress and knock down tRFs in *Drosophila*. Because *tRF-Trp-CCA-014* is conserved in flies, mice, and humans, mouse NIH3T3 cells were used as the model to research the function of *tRF-Trp-CCA-014*. The fat body of fruit flies is equivalent to the mammalian liver [21]. Thus, *tRF-Trp-CCA-014* has a similar expression pattern in tissues, such as higher expression levels in the head and fat body of aged fruit flies at 42 days and in the brain and liver of aged mice. On the other hand, a previous study reported that the intersection of the altered transcriptomes revealed 50 RNAs consistently elevated and 18 RNAs consistently reduced across four senescence models [35]. Thus, not all genes maintain the same expression pattern during the ageing process. In the present study, five tRFs were detected in aged mouse NIH3T3 cells and only *tRF-Trp-CCA-014* had the same expression trend as that in aged fruit flies. These results suggest that *tRF-Trp-CCA-014* has tissue specificity and stage specificity. Additionally, the mimics and inhibitors of *tRF-Trp-CCA-014* did not induce senescence of mouse NIH3T3 cells (Appendix A). Thus, *tRF-Trp-CCA-014* may accumulate during the ageing process of mouse NIH3T3 cells and have no effect on cell senescence. Therefore, we speculate that *tRF-Trp-CCA-014* may be produced by ADR-stimulated mouse NIH3T3 cells and it may accumulate during the ageing process of mouse NIH3T3 cells and *Drosophila* and have no effect on senescence.

Based on its expression level and patterns of *tRF-Trp-CCA-014*, it may be involved in other biological processes. RNA-seq was used to analyse the DEGs when the mimics and inhibitors of *tRF-Trp-CCA-014* were transfected into mouse NIH3T3 cells. The number of DEGs was screened and 26 functional DEGs were enriched in three differential signalling pathways (immune system, cancer: overview, and immune disease). Thus, the targets of *tRF-Trp-CCA-014* were identified from those DEGs based on the miRNA-like mode [36]. The mimics and inhibitors of *tRF-Trp-CCA-014* could affect the *H3C4* gene mRNA and protein levels in mouse NIH3T3 cells. *tRF-Trp-CCA-014* was shown to bind to the *H3C4* 3′UTR in a dual-luciferase reporter gene assay. In addition, *tRF-Trp-CCA-014* and *H3C4* were detected in the cytoplasm of aged NIH3T3 cells. These results suggest that the *H3C4* gene is the target of *tRF-Trp-CCA-014*.

Histones, as basic nuclear proteins, are an important nucleosome structure of the chromosomal fibre in eukaryotes [37]. The *H3C4* gene is a member of the histone H3 family and encodes a replication-dependent histone. Previous studies have reported that *H3C4* is involved in the epigenetic regulation of gene expression [38], nucleosome assembly [39], and telomere organization [40]. Previous studies found that telomeres can maintain chromosomal stability and are associated with ageing and longevity [41,42]. Although *tRF-Trp-CCA-014* did not affect cell senescence, it may regulate the expression level of H3C4 mRNA and protein to function in ageing; otherwise, the *H3C4* isoforms of human, mouse, and fruit fly are conserved. Therefore, *tRF-Trp-CCA-014* may regulate the expression of the *H3C4* gene to affect related functions and signalling pathways, which deserves further study in our future work.

## 4. Materials and Methods

### 4.1. tsRNAs Analysis

Our published raw data (NCBI SRA_PRJNA716466) [43] were used to analyse tsRNA expression patterns at 7 days and 42 days. Library preparation and sequencing with standard small RNA-Seq protocols was performed to obtain high-quality data. In the sequencing process, m1A and m3C demethylation for efficient reverse transcription can guarantee maximum enrichment of tsRNAs in *Drosophila*. Sequencing quality was examined by FastQC (http://www.bioinformatics.babraham.ac.uk/projects/fastqc/ (accessed on 1 January 2018)), and trimmed reads (pass Illumina quality filter, trimmed 5′, 3′-adaptor bases by cutadapt [44]) were aligned, allowing for only 1 mismatch to the mature tRNA sequences. Then, reads that did not map were aligned, allowing for only 1 mismatch to precursor tRNA sequences with bowtie software [45]. The abundance of tsRNAs was evaluated using their sequencing counts and was normalized as counts per million of total aligned reads (CPM). The differentially expressed tsRNAs were screened based on the count value with the R package edgeR [46]. Principal component analysis (PCA), correlation analysis, pie plots, Venn plots, hierarchical clustering, scatter plots, and volcano plots were performed in the R or Perl environment for statistical computing and generation of graphics of the expressed tsRNAs.

### 4.2. GO and KEGG Enrichment

Gene ontology (GO) enrichment was applied to analyse biological processes (BP), cellular components (CC), and molecular functions (MF) via the GOseq R package [47]. To screen significant signalling pathways, Kyoto Encyclopedia of Genes and Genomes (KEGG) enrichment analysis was performed using the clusterProfiler (version 3.16.0) package (Bioconductor, Guangzhou, China) in the R environment [48]. A corrected *p* value less than 0.05 was considered to indicate a significantly enriched term for the GO and KEGG analyses.

### 4.3. qPCR

Total RNA was extracted using TRIzol reagent (Invitrogen, Waltham, MA, USA) and quantified by a NanoDrop-1000 instrument. Then, first-strand cDNA was synthesized by two methods. First, first-strand cDNA was synthesized by RT Easy TM II (with gDNase) (FOREGENE, Chengdu, China), which was utilized to test the expression level of mRNA genes. Second, first-strand cDNA was produced by an MR101-01 miRNA first-strand cDNA synthesis kit (by stem-loop) (Vazyme, Nanjing, China), which was prepared to determine the expression level of tsRNAs. Real-time quantitative PCR (qPCR) was routinely applied to measure relative gene expression levels using the TB Green Fast qPCR mix (Takara, Kyoto, Japan). All operations followed the manufacturer′s instructions for the kits. 2-ΔΔCT [49] was performed to identify the mRNA gene and tRFs expression levels according to the cycle threshold values. The relative levels of mRNA genes and tRFs were normalized to *ribosomal protein L32* (*RP49*) and small nuclear RNA *U6*, respectively. Differential expression levels were compared by the IBM SPSS Statistics 20 system at a significance level of less than 0.05 [50]. The stem-loop and qPCR primers are summarized in Table 1.

### 4.4. Cell Culture and Cell Senescence Model

Mouse embryonic fibroblast NIH/3T3 cells were provided by our laboratory and used to build the senescence model. NIH/3T3 cells were cultured in DMEM (Dulbecco’s modified Eagle’s medium) (ATCC, Cambridge, MA, USA) supplemented with 10% foetal bovine serum (ThermoFisher Scientific, Waltham, MA, USA) and 1% penicillin/streptomycin solution (Sangon Biotech, Shanghai, China) in a 37 °C, 5% CO_2_ incubator. Cell transfection was performed according to the LipoMaxTM transfection reagent (Sudgen, Nanjing, China) instruction manual, and the transfection concentrations of mimics, inhibitors, and negative control were all 20 pmol/μL. NIH/3T3 cells in their exponential growth phase were treated with 0.5 μM adriamycin (ADR) (Merck, Darmstadt, Germany). After 72 h, the cells were collected and used to confirm cellular senescence. The mimics, inhibitors, and negative control of *tRF-Trp-CCA-014* were designed and synthesized by Sangon Biotech Co., Ltd. Shanghai, China.

### 4.5. SA-β-gal Staining

Senescent mouse NIH/3T3 cells were collected and stained blue by *β*-galactosidase staining agent (Beyotime, Shanghai, China) to determine whether the cells were senescent. The experiment was performed according to the instructions. After staining, the senescent NIH3T3 cells treated with ADR were obviously stained blue and the cells became larger and had an irregular shape.

### 4.6. Nuclear Cytoplasmic Separation

To confirm the expression location of *tRF-Trp-CCA-014* in mouse NIH/3T3 cells, the nucleus and cytoplasm were separated by a cytoplasmic and nuclear RNA purification kit (Norgen Biotek, Mississauga, Canada) and used to extract RNA. Then, qPCR and 2-ΔΔCT [23] were utilized to identify the *tRF-Trp-CCA-014* expression level according to the cycle threshold values. The small nuclear RNA *U6* and *glyceraldehyde 3-phosphate dehydrogenase* (*GAPDH*) were used as the reference genes in the nucleus and cytoplasm of cells, respectively.

### 4.7. Tissue Expression Specificity of tRF-Trp-CCA-014

The relative expression level of *tRF-Trp-CCA-014* was detected in the *Drosophila* samples of egg, 1st stage larva (L1), 2nd stage larva (L2), 3rd stage larva (L3), and pupa (P) and adults of 7 days (7 d), 14 days (14 d), 21 days (21 d), 28 days (28 d), 30 days (30 d), 35 days (35 d), 42 days (42 d), and 49 days (49 d) of age, and different tissues (head, fat body, gut, and ovary) from 42 d by qPCR method. Furthermore, the relative expression level of *tRF-Trp-CCA-014* was also determined in 5-month-old C57 wild-type young mice and 19-month-old C57 wild-type aged mice. All mice were purchased from Nanjing Junke Bioengineering Co., Ltd., China. The tissues included thigh muscle, liver, brain, lung, gut, heart, kidney, spleen, midbrain, and cerebellum. All samples had three biological replicates.

### 4.8. RNA-seq

RNA-seq technology was used to study the transcriptome of mouse NIH3T3 cells. The cells treated for 48 h contained six groups, including Mimics, Inhibitors, Mimic NC, Inhibitor NC, Aging (ADR), and Control (Normal). Each sample contained three biological replicates. RNA isolation, cDNA library construction, and sequencing were described in a previous study [51]. Hisat2 [52] was utilized to align the clean reads (100 bp paired-end reads) with the reference genome (*Mus musculus* GRCm39) to obtain the position information on the reference genome or gene, as well as the unique sequence feature information of the sequenced sample. Then, the known reference gene sequences and annotation files from databases were used for sequence similarity alignment to identify the expression abundance of each protein-coding gene in each sample. The htseq-count [53] was utilized to obtain the number of reads aligned to the protein-coding genes in each sample. After the counts were obtained by comparison, protein-coding genes were filtered to remove genes with zero reads.

Cuffdiff (v2.1.1) [54] was used to calculate FPKMs (fragments per kilobase of transcript sequence per millions base pairs sequenced) of coding genes in each sample. The FPKM method can eliminate the influence of differences in protein-coding gene length and sequencing quantity on the calculated protein-coding gene expression. The differentially expressed genes (DEGs) were identified as those with a corrected *p* value ≤ 0.05 (using BH orthosis) and fold change > 1.5. Furthermore, GO and KEGG functional enrichment analyses of DEGs were performed.

### 4.9. Target Prediction of tRF-Trp-CCA-014

The target genes of *tRF-Trp-CCA-014* were screened in two ways. The first was identification based on DEGs (Aging (ADR) vs. Control (Normal) group, Mimics vs. Inhibitors group) that were enriched in the GO and KEGG analyses. The second was identification based on DEGs (Mimics vs. Inhibitors group) from transcription factor families. Then, RNAhybrid [55] was utilized to predict the binding sites between *tRF-Trp-CCA-014* and target genes.

### 4.10. Dual-Luciferase Reporter Gene Assay

Human 293T/17 cells were plated in 96-well culture plates and transfected with the mimics of *tRF-Trp-CCA-014* and target gene plasmid, the negative control and the mimics of the target gene, the mutants of target gene and the mimics of *tRF-Trp-CCA-014*, and the mutants of target gene and the mimics of the negative control. The Renilla luciferase expression vector pRL-TK (Promega, Madison, WI, USA) and Lipofectamine™ 2000 transfection reagent (Invitrogen, Waltham, MA, USA) were used in this experiment. Next, a dual-luciferase system (Promega) was utilized to assay the cells for both firefly and Renilla luciferase activities according to the manufacturer’s protocol.

### 4.11. Western Blot

The H3C4 protein expression level was detected by Western blotting. Total proteins were obtained from mouse NIH/3T3 cells using a ProteoPrepÒ sample extraction kit (Merck, Darmstadt, Germany) and a BCA protein assay kit (Merck, Germany) was used to measure the protein concentrations. First, the total protein was blotted onto polyvinylidene fluoride membranes. Then, the membranes were incubated at 4 °C overnight with primary antibodies (Proteintech, Chicago, IL, USA) against H3C4 and GAPDH. Next, the membranes were incubated with HRP-conjugated secondary antibodies. GraphPad Prism 9 was used to analyse the H3C4 level based on the grey value of the protein band.

### 4.12. RNA in Situ Hybridization

Aged mouse NIH3T3 cells were used to analyse the location of *tRF-Trp-CCA-014* and its target genes. First, the probes were designed and synthesized according to the sequences of *tRF-Trp-CCA-014* and its target genes. Then, the cells were treated with 4% paraformaldehyde and the cell membranes were permeabilized by Triton-100 (Beyotime, Shanghai, China) for 5 min at 4 °C. Next, the remaining steps were performed according to Xu et al. (2021) [56]. Finally, photographs were taken using a fluorescence microscope.

### 4.13. Statistical Analysis

GraphPad Prism software was used to statistically analyse the experimental results. The differences in experimental data among groups were compared by single-factor ANOVA. *, **, and *** indicate *p* < 0.05, *p* < 0.01, and *p* < 0.001, respectively.

## 5. Conclusions

In conclusion, the expression profiles of tsRNAs in *Drosophila* were analysed at 7 days and 42 days and the conserved tRF *tRF-Trp-CCA-014* was identified in our study. Then, the essential function of *tRF-Trp-CCA-014* was predicted in mouse NIH3T3 cells and the binding properties of the *tRF-Trp-CCA-014*-*H3C4* network were validated. These results will provide the foundation for the study of tsRNAs in *Drosophila* and mice. However, unlike lncRNAs and circRNAs, there are no research methods to study the function and mechanism of specific tsRNAs in vivo, such as genetic techniques or biochemical methods. Developing such methods is still greatly challenging because of the short sequences of tsRNAs. Thus, future studies should explore effective methods to identify gain-of-function and loss-of-function mutations of tsRNAs in vivo.

## Figures and Tables

**Figure 1 ijms-24-06169-f001:**
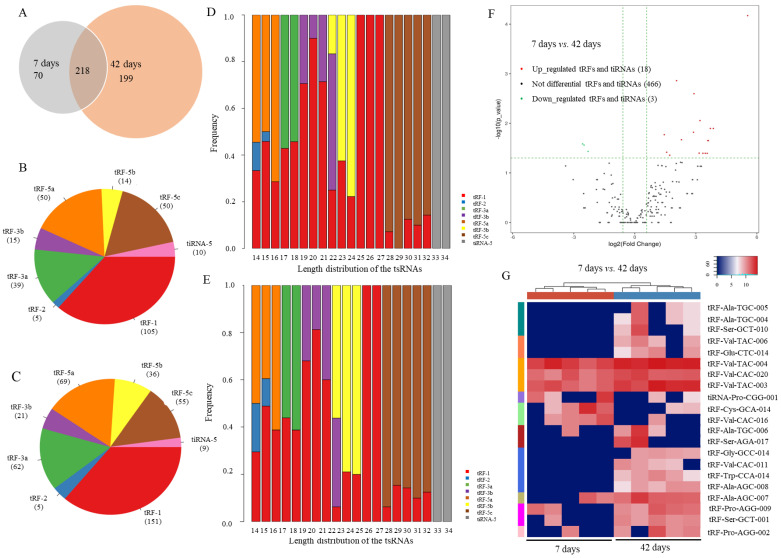
Expression pattern of tsRNAs in *Drosophila* at 7 days and 42 days. (**A**) newly discovered and known tsRNAs; (**B**) classification of tsRNAs at 7 days; (**C**) classification of tsRNAs at 42 days; (**D**) subtype length of tsRNAs at 7 days; (**E**) subtype length of tsRNAs at 42 days; (**F**) significantly up/downregulated tsRNAs between 7 days and 42 days; (**G**) heatmap of 21 differentially expressed tsRNAs.

**Figure 2 ijms-24-06169-f002:**
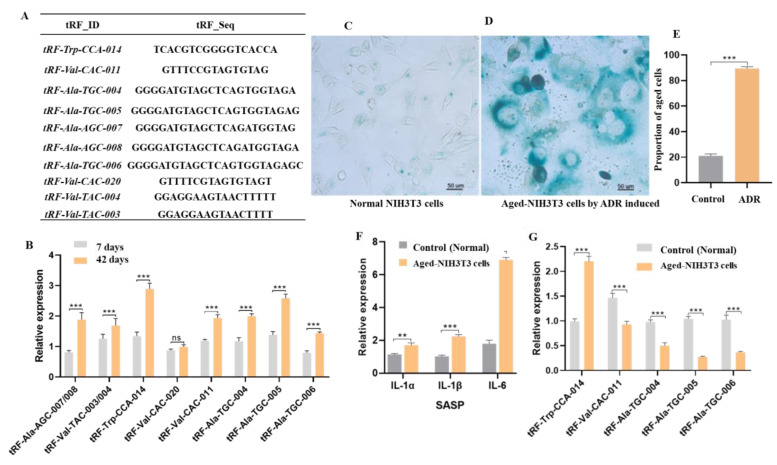
Verification of 10 differentially expressed tRFs in *Drosophila* and aged mouse NIH3T3 cells. (**A**) The sequences of 10 differentially expressed tRFs; (**B**) the expression levels of 10 differentially expressed tRFs of *Drosophila* at 7 days and 42 days; (**C**) β-gal staining of normal mouse NIH3T3 cells without adriamycin (ADR); (**D**) β-gal staining of aged mouse NIH3T3 cells induced by ADR; (**E**) the number of β-gal-stained cells in normal and aged mouse NIH3T3 cells. (**F**) the expression levels of *IL-1α*, *IL-1β*, and *IL-6*; (**G**) 5 tRFs in normal and aged mouse NIH3T3 cells induced by ADR. ADR—adriamycin; SASP—senescence-associated secretory phenotypes; **, *p* < 0.01; ***, *p* < 0.001; ns—no significance.

**Figure 3 ijms-24-06169-f003:**
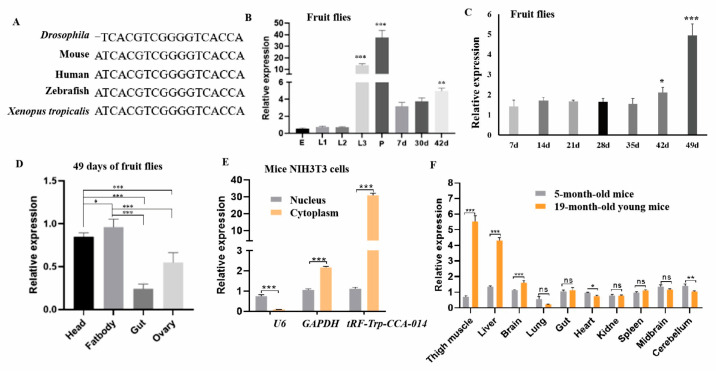
Expression features of *tRF-Trp-CCA-014* in *Drosophila* and mice. (**A**) The sequence alignment of *tRF-Trp-CCA-014* in *Drosophila*, mouse (a relational database of transfer RNA related fragments 3002a, tRFdb 3002a), human (from MINTbase v2.0), zebrafish (tRFdb 3038), and *Xenopus tropicalis* (tRFdb 3015); (**B**) the expression level of *tRF-Trp-CCA-014* in larvae and adults; (**C**) the expression level of *tRF-Trp-CCA-014* in more adult ages; (**D**) the expression level of *tRF-Trp-CCA-014* in 49-day-old fruit flies; (**E**) the expression level of *tRF-Trp-CCA-014* in the nucleus and cytoplasm of mouse NIH3T3 cells; (**F**) the expression level of *tRF-Trp-CCA-014* in different tissues of mice. E—egg; L1—the first instar larva; L2—the second instar larva; L3—the third instar larva; P—pupa; d—days. *, *p* < 0.05; **, *p* < 0.01; ***, *p* < 0.001; ns—no significance.

**Figure 4 ijms-24-06169-f004:**
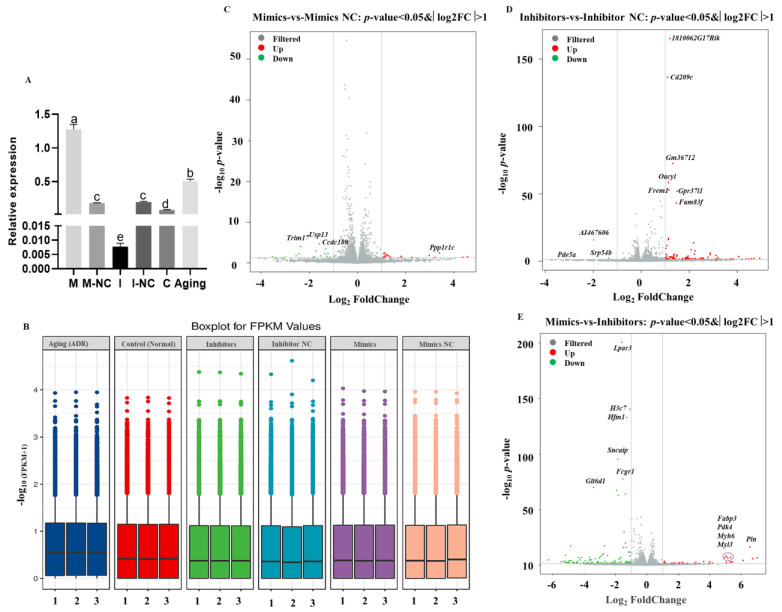
Differentially expressed genes (DEGs) identified by RNA-seq after transfection of *tRF-Trp-CCA-014* in mouse NIH3T3 cells. (**A**) The expression level of *tRF-Trp-CCA-014* in Mimics, Mimic NC, Inhibitors, Inhibitor NC, Aging (adriamycin (ADR)), and Control (Normal) groups; (**B**) boxplot for FPKM values in different groups; (**C**–**E**) the significantly up/downregulated DEGs among Mimics vs. Mimic NC groups, Inhibitors vs. Inhibitor NC groups, and Mimics vs. Inhibitors groups. In (**A**), different small letters indicate significant differences at *p* < 0.05 level.

**Figure 5 ijms-24-06169-f005:**
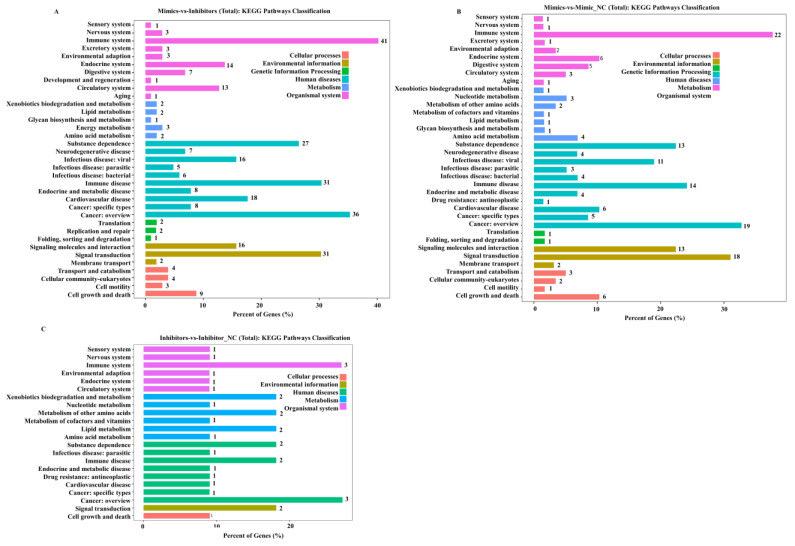
KEGG enrichment analyses of differentially expressed genes (DEGs) identified by RNA-seq in mouse NIH3T3 cells transfected with *tRF-Trp-CCA-014* (*p* < 0.05). (**A**–**C**) KEGG signalling pathways of Mimics vs. Inhibitors groups, Mimics vs. Mimic NC groups, and Inhibitors vs. Inhibitor NC groups, respectively. NC—negative control.

**Figure 6 ijms-24-06169-f006:**
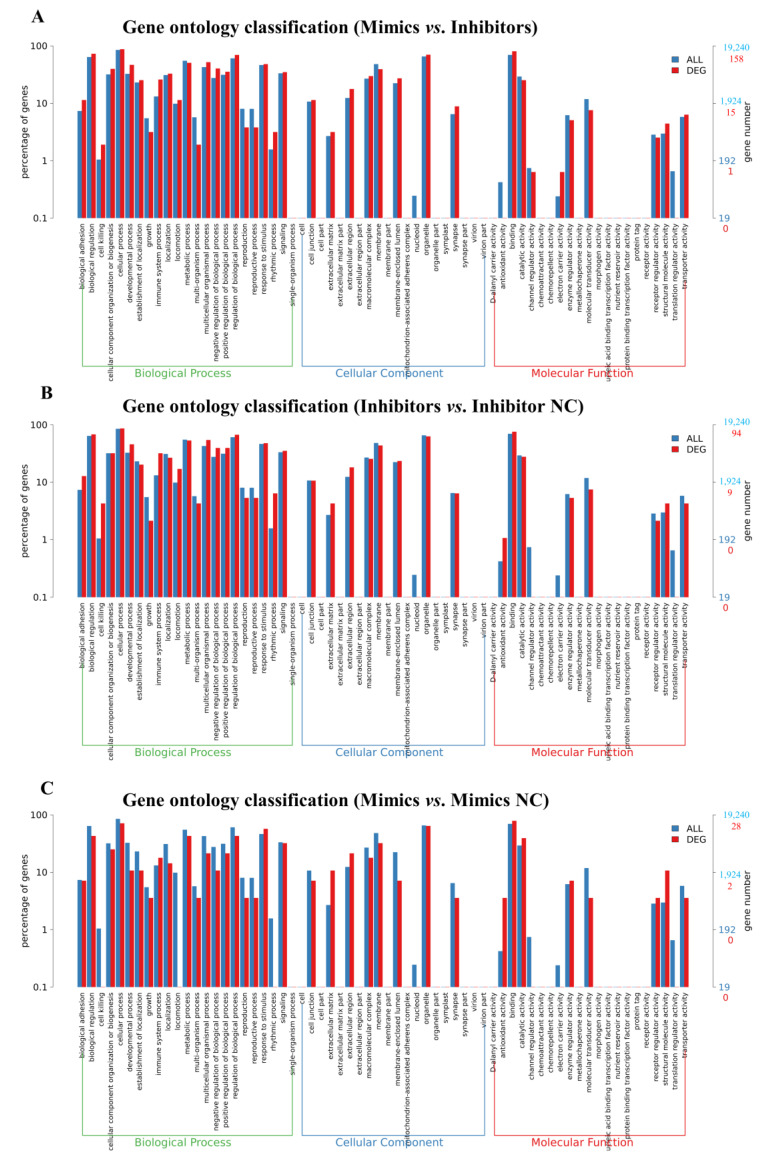
GO enrichment analyses of differentially expressed genes (DEGs) identified by RNA-seq in mouse NIH3T3 cells transfected with *tRF-Trp-CCA-014* (*p* < 0.05). (**A**–**C**) GO terms of Mimics vs. Inhibitors groups, Inhibitors vs. Inhibitor NC groups, and Mimics vs. Mimic NC groups, respectively. NC—negative control.

**Figure 7 ijms-24-06169-f007:**
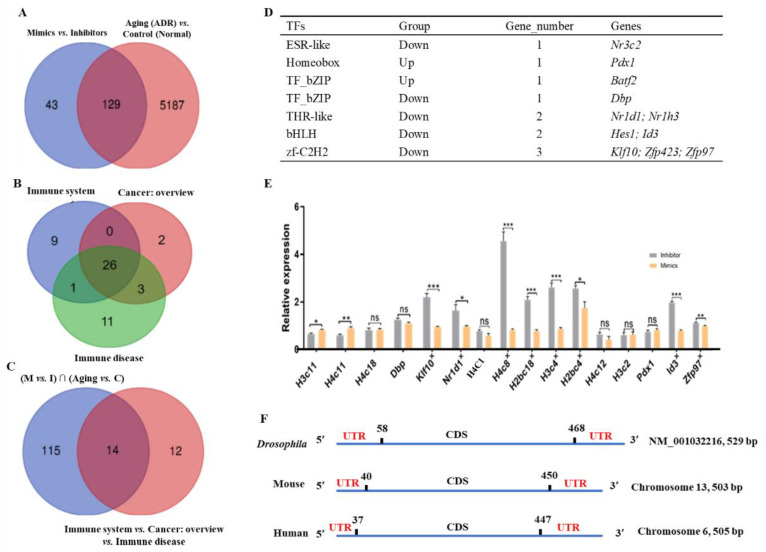
Target screening of *tRF-Trp-CCA-014* in mouse NIH3T3 cells. (**A**) the overlapping DEGs between Mimics vs. Inhibitors groups and Aging (adriamycin ageing (adriamycin (ADR)) vs. Control (Normal) groups; (**B**) the overlapping DEGs in the immune system, cancer: overview, and immune disease; (**C**) the overlapping DEGs in (**A**,**B**); (**D**) the differentially expressed transcription factors between the Mimics vs. Inhibitors groups, including two upregulated genes and nine downregulated genes in the Inhibitors group when compared with the Mimics group; (**E**) the relative expression level of DEGs by qPCR; (**F**) sequence alignment of the *H3C4* gene in mouse, human, and *Drosophila*. In (**C**): M—mimics; I—inhibitors; Aging—with ADR treatment; C—control (without ADR treatment); ∩—intersection in mathematics. UTR—untranslated region; CDS—coding sequence; *, *p* < 0.05; **, *p* < 0.01; ***, *p* < 0.001; ns—no significance.

**Figure 8 ijms-24-06169-f008:**
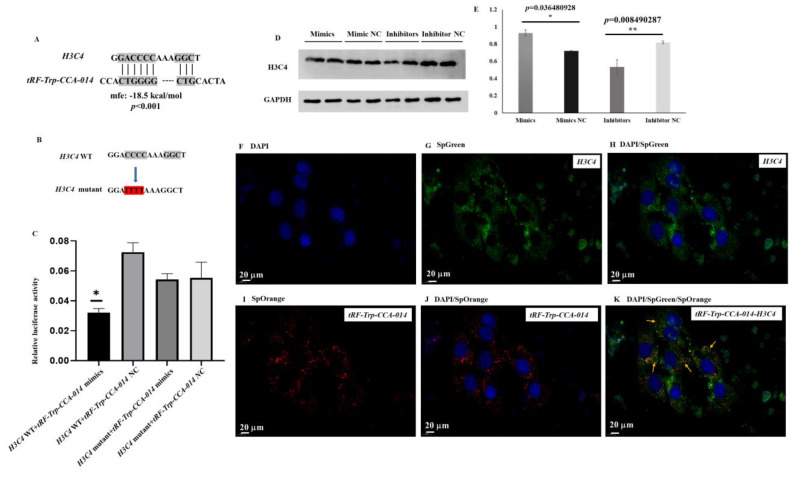
Analysis of the *tRF-Trp-CCA-014*-*H3C4* network in mouse NIH3T3 cells. (**A**) Binding sites between *tRF-Trp-CCA-014* and *H3C4*; (**B**) mutant sites of the *H3C4* 3′UTR; (**C**) binding properties of *tRF-Trp-CCA-014*-*H3C4* by dual luciferase reporter gene assay in 293T cells. (**D**) H3C4 protein expression level after transfection with *tRF-Trp-CCA-014* mimics, negative control of *tRF-Trp-CCA-014* mimics, *tRF-Trp-CCA-014* inhibitors, and negative control of *tRF-Trp-CCA-014* inhibitors in mouse NIH3T3 cells by Western blot. (**E**) quantification of protein bands in Figure 7D; (**F**) DAPI; (**G**) SpGreen for *H3C4*; (**H**) merged fluorescent images of (**F**,**H**); (**I**) SpOrange for *Trp-CCA-014*; (**J**) merged fluorescent images of (**F**,**I**); (**K**) merged fluorescent images of (**F**,**G**,**I**). NC—negative control; UTR—untranslated region; *, *p* < 0.05; **, *p* < 0.01.

**Table 1 ijms-24-06169-t001:** Primer sequences for real-time quantitative PCR (qPCR).

Gene Name	Forwards Primers	Reverse Primers
*tRF-Trp-CCA-014*	GGCGGTCACGTCGGGG	AGTGCAGGGTCCGAGGTATT
*tRF-Val-CAC-020*	GCGCGCGGTTTTCGTAG	AGTGCAGGGTCCGAGGTATT
*tRF-Val-TAC-003*	GCGCGCGGAGGAAGTA	AGTGCAGGGTCCGAGGTATT
*tRF-Val-TAC-004*	GCGCGCGGAGGAAGTAA	AGTGCAGGGTCCGAGGTATT
*tRF-Ala-AGC-007*	GCGGGGGATGTAGCTCAGA	AGTGCAGGGTCCGAGGTATT
*tRF-Ala-AGC-008*	GCGGGGGATGTAGCTCAGAT	AGTGCAGGGTCCGAGGTATT
*tRF-Val-CAC-011*	GCGCGCGGTTTCCGTA	AGTGCAGGGTCCGAGGTATT
*tRF-Ala-TGC-004*	GCGGGGGATGTAGCTCAGT	AGTGCAGGGTCCGAGGTATT
*tRF-Ala-TGC-005*	CGGGGGATGTAGCTCAGTG	AGTGCAGGGTCCGAGGTATT
*tRF-Ala-TGC-006*	CGGGGGATGTAGCTCAGTGG	AGTGCAGGGTCCGAGGTATT
*U6*	TCTTGCTTCGGCAGAACATA	GATTTTGCGTGTCATCCTTG
*RP49*	GCCCAAGGGTATCGACAACA	CTTGCGCTTCTTGGAGGAGA
*GAPDH*	TCTTCCAGGCGAACCACTTC	AAGACGTCCACGAAGCGAAT
*H3C11*	GTGAAGAAGCCTCACCGCTA	TACGTGCCAGTGTAGAAGGC
*H4C11*	TGCTCCATAGCCATGTCTGG	TTGGTGATGCCCTGGATGTT
*H4C18*	ATCTCCGGCCTCATCTACGA	TAATTAGCCGCCGAATCCGT
*Dbp*	GCAGAGTCCTGTTCCTTGCT	ATTGTGTTGATGGAGGCGGT
*Klf10*	CACCAGTGTCATCCGTCACA	GGCTCAGGCTTGGATCTGTT
*Nr1d1*	GAAGTGTCTCTCCGTTGGCA	GAAGTGTCTCTCCGTTGGCA
*H4C1*	ACTGGCTAGTGAGCTTCCTT	TCGTAGATGAGGCCGGAGAT
*H4C8*	ATCTCCGGCCTCATCTACGA	AAGGGCCTTTTGTAGCAGAAAT
*H2bC18*	ATCACTTCCCGGGAGATCCA	AGCCTTTTGGGTAAAGCCGA
*H3C4*	GCCTACCTTGTGGGTCTGTT	AAGAGCCTTTGGTTAATTCCGT
*H2bC4*	TACAACAAGCGCTCGACCAT	GGAATTCGCTACGGAGGCTT
*H4c12*	GTGAACGACATCTTCGAGCG	GGTGCTAGACGTCAACCCT
*Pdx1*	AGCGTTCCAATACGGACCAG	TGCTCAGCCGTTCTGTTTCT
*H3c2*	GTTGCTTGTTTCTACCATGCCC	TCGAGCGCTTGTTGTAATGC
*Id3*	CTGAAGAGCTAGCACACGCT	CTCTCGACACCCCATTCTCG
*Zfp97*	TCCGGAATCCTTTTCGCTGG	TGCCTGAGCTTCCTTCACAG
*IL-1α*	TCTCAGATTCACAACTGTTCGTG	AGAAAATGAGGTCGGTCTCACTA
*IL-1β*	GAAATGCCACCTTTTGACAGTG	TGGATGCTCTCATCAGGACAG
*IL-6*	ATCCAGTTGCCTTCTTGGGACTGA	TAAGCCTCCGACTTGTGAAGTGGT

## Data Availability

The data presented in this study are available on request from the corresponding author without any restrictions.

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
