# Peer review of "The Expression Pattern of tRNA-Derived Small RNAs in Adult Drosophila and the Function of tRF-Trp-CCA-014-H3C4 Network Analysis"

_ijms, 2023, doi:10.3390/ijms24076169_

Round 1

Reviewer 1 Report

Yang et al aims to analyze the expression pattern of tRNA-derived small RNAs (tsRNAs) in aging Drosophila and the function of a specific tRF, which is overexpressed in aging Drosophila and mice. To this end, authors first obtain the small RNA-seq data from 7-day- and 42-day-old flies. They validate the expression of tRF-trp-CCA-014 both in aging Drosphila and mice. They then generate an adriamycin-mediated aging model in mouse NIH3T3 cells. By using these cells, they carry out a transcriptome study with the cells transfected with the mimics and inhibitors of tRF-trp-CCA-014, along with the cells induced with adriamycin. The authors examine the biological processes impacted by the transfection of mimics or inhibitors. They also identify a target gene. Their findings show that tRF-trp-CCA-014 is upregulated in aging Drosophila and mice. Although the overexpression of this tRF does not result in aging, it appears to have a miRNA-like function by targeting H3C4 gene.

Although there are several transcriptome data that present the expression pattern of tRFs in different developmental stages of Drosophila, a complete analysis in an aging model is lacking. Thus, I believe that the findings presented in this manuscript could be useful to the scientists working in this field. However, I believe that the following changes are required before it can be considered for publication. 

Major points:

Abstract:

1. Please revise the abstract to improve clarity. Although the title does not say anything about mice, the majority of work is conducted with a mouse cell line. Please avoid sudden transitions between sentences.

2. line 35, please remove “human”. I did not see any data associated with human cells.

Results:

1. I did not have any access to Datasets 1-4. Either they were not included in the submission or not made available for reviewers.

2. Figure 1: In Panels B-C, there are two tRF-5a. Probably mislabeling. Please correct.

3. Figure 2: (a) Line 125: Please explain why the expression levels of these cytokines were measured. Probably to validate the effect of adriamycin. (b) Lines 126-127, Figure 2G: The expression patterns of other tRFs are contradictory between Drosophila and mice. How to explain this. Also, is it possible that the overexpression of tRF-Trp-CCA-014 could be because of yet another effect of adriamycin, rather than aging? Please at least include a statement raising this possibility.

4. Figure 3: The lowercases used in this figure as well as other figures are confusing. Please either remove them or precisely explain what each lowercase letter stands for.

5. Figure 4: (a) The authors use synthetic mimics and inhibitors to delineate the potential function(s) of tRF-Trp-CCA-014. Considering the fact that tRNAs, and tRFs as a result, are chemically modified, I wonder whether the authors considered the effect of the absence of modifications on the functionality of synthetic tRFs. (c) line 176, please include the data on the effect of tRF-Trp-CCA-014 on senescence.

6. Figure 5: Panel C and F are mislabeled. What is shown on the figure does not match the caption. Please correct.

7. Figure 7: (a): The sequence of wt gene in Panel A is different from that in Panel B. Which one is correct? (also correct lines 252-253, accordingly). (b) What is shown in the graph in Panel E is not supported by the data shown in the western blot (Panel D). I do not quite agree that mimic overexpression downregulates its target based on the western blot. Also, inhibitor appers to downregulate the target protein amount, which is stated just the opposite in lines 260-261. I am affraid that I am not convinced with what is presented in Panels A-E.

Discussion:

1. The authors should provide some explanation as to how tRF-Trp-CCA-014-mediated regulation of H3C4 could lead to aging.

2. The authors shold benefit from the existing Drosophila small RNA-seq data, such as those obtained from embryos. Please discuss the expression pattern of tRFs in embryos and adults to put things in a developmental perspective (See Goktas et al 2017, PMID: 29156628 and Cosacak et al 2018, PMID: 29439397).

Methods:

1. Lines 357-358: Please remove the human cell line. I did not see any data associted with this cell line being presented in the manuscript.

2. Please deposit the RNA-seq data to a public database such as GEO and include their GEO IDs in the manuscript.

Also, the manuscript could benefit greatly from revision by a professional editor.

Minor points:

1. Line 19, “RNAs Illumina sequencing” should be “RNA sequencing”

2. Please replace all “&” with “and”.

Author Response

Dear reviewers,

We would like to thank you for the kind letter and constructive comments concerning our article (Submission ID ijms-2255041). All these comments are valuable and helpful for improving our article. The authors have seriously discussed all these comments. According to the suggestions, we have done our efforts to revise the manuscript to meet the requirements of your journal. All modifications in the revised manuscript are marked in red color. The followings are our point-by-point responses.

Yours Sincerely,

Yang Deying

Reviewer 2 Report

This is a nice work on one of the weirdest class of eukaryotic RNAs, those derived from tRNAs. Authors sequenced tsRNAs of Drosophila from young (7 days old insects) and old (42 days old insects) and performed an expression analysis as well as a GO/KEGG analysis. The suggestion that tRFs could work as miRNAs is interesting In general this is a well performed work but the way results are presented is very poor. Lettering is too small in most of the figures, making it difficult to follow the results.

Concerns

-Figure 1,B and C. The names of the tRFs and tiRNAs are too small to be seen. In general this is a problem of ALL the figures. I suggest authors to re-letter all the figures with bigger fonts.

-I would suggest that the volcano-plots (Fig 1F; Figure 4) named the significantly up/downregulated RNAs.

-In lines 223 and 226 DGEs are mistakenly written instead of DEGs

-The characterization of tRF targets is very difficult to follow. These data should be presented in a new figure explaining HOW these genes were selected.

-Figure 7 is very difficult to follow. Plates are underexposed and these can hardly be seen. Not to say that scale bars are barely visible. Arrows/arrowheads should be used to point features of interest.

-Are there other targets of tRF-Trp-CCA-014 other than H3C4?  Authors should look for pieces of homology among the 3’UTR of H3C4 and other transcripts.

-The English language is very poor. Is it strongly recommended that the authors revise the text with an English-speaking native .

Author Response

(The authors gave the same response as above.)

Round 2

Reviewer 1 Report

The authors have addressed all the points that I have raised in the previous round of review. The revisions are sufficient to warrant publication at the current form of the manuscript. 

One minor point: Figure 5: In the caption, there are panels 5D-F, but the corresponding data is labeled as D. Please either label the data as D-F or change the caption accordingly.

Author Response

Response to Reviewer 1

One minor point: Figure 5: In the caption, there are panels 5D-F, but the corresponding data is labeled as D. Please either label the data as D-F or change the caption accordingly.

Response: Thanks for your suggestions. The labels in the Figure 5 and legend have been revised.

Reviewer 2 Report

Although I acknowledge the efforts made by the authors to improve the manuscript, I think that there's still room for further improvement.

-In Figure 4D/E there are a few genes that are extremely up/down-regulated and I suggest that the authors identify them. Just writing the names in the own figure would be OK.

-The name of the gene groups at the left of Figure 5A/B/C  is still very difficult to read (not to say those at the bottom of in Figure 5D) even at a 200% magnification. I would suggest the authors to split tis figure in two  and increase font size to be read at the normal 125% magnification

Author Response

Response to Reviewer 1

Point 1: -In Figure 4D/E there are a few genes that are extremely up/down-regulated and I suggest that the authors identify them. Just writing the names in the own figure would be OK.

Response: Thanks for your constructive suggestions. The names of extremely up/down-regulated genes have been added to the Figure 4 C/D/E.

Point 2: -The name of the gene groups at the left of Figure 5A/B/C is still very difficult to read (not to say those at the bottom of in Figure 5D) even at a 200% magnification. I would suggest the authors to split tis figure in two and increase font size to be read at the normal 125% magnification.

Response: It has been divided into Figure 5 and Figure 6 in the revised manuscript.